# Detection of Porphyrins in Hair Using Capillary Liquid Chromatography-Mass Spectrometry

**DOI:** 10.3390/ijms23116230

**Published:** 2022-06-02

**Authors:** Marwa Louleb, Ismael Galván, Latifa Latrous, Nicholas M. Justyn, Geoffrey E. Hill, Ángel Ríos, Mohammed Zougagh

**Affiliations:** 1Regional Institute for Applied Scientific Research, IRICA, 13005 Ciudad Real, Spain; marwa.lauleb@gmail.com (M.L.); angel.rios@uclm.es (Á.R.); 2Laboratory of Applied Mineral Chemistry (LCMA) LR19ES02, Department of Chemistry, Faculty of Sciences of Tunis, Campus Universitaire Farat Hached El Manar 1, University of Tunis El Manar, Tunis 2092, Tunisia; latifa.latrous@ipeiem.utm.tn; 3Department of Analytical Chemistry and Food Technology, University of Castilla—La Mancha, 13071 Ciudad Real, Spain; 4Department of Evolutionary Ecology, National Museum of Natural Sciences, CSIC, 28006 Madrid, Spain; 5Department of Biological Sciences, Auburn University, Auburn, AL 36849, USA; nmj0005@auburn.edu (N.M.J.); hillgee@auburn.edu (G.E.H.); 6Analytical Chemistry and Food Technology Department, Faculty of Pharmacy, University of Castilla—La Ancha, 02008 Albacete, Spain

**Keywords:** animal pigmentation, fluorescence, hair, mammals, natural porphyrins, CLC-MS, flying squirrels

## Abstract

Unlike humans, some animals have evolved a physiological ability to deposit porphyrins, which are pigments produced during heme synthesis in cells, in the skin and associated integument such as hair. Given the inert nature and easiness of collection of hair, animals that present porphyrin-based pigmentation constitute unique models for porphyrin analysis in biological samples. Here we present the development of a simple, rapid, and efficient analytical method for four natural porphyrins (uroporphyrin I, coproporphyrin I, coproporphyrin III and protoporphyrin IX) in the Southern flying squirrel Glaucomys volans, a mammal with hair that fluoresces and that we suspected has porphyrin-based pigmentation. The method is based on capillary liquid chromatography-mass spectrometry (CLC-MS), after an extraction procedure with formic acid and acetonitrile. The resulting limits of detection (LOD) and quantification (LOQ) were 0.006–0.199 and 0.021–0.665 µg mL^−1^, respectively. This approach enabled us to quantify porphyrins in flying squirrel hairs at concentrations of 3.6–353.2 µg g^−1^ with 86.4–98.6% extraction yields. This method provides higher simplicity, precision, selectivity, and sensitivity than other methods used to date, presenting the potential to become the standard technique for porphyrin analysis.

## 1. Introduction

Porphyrins are pyrrole-based macrocycles in which the heteroaromatic unities are bridged by methylene or methine groups (Table 1). These compounds include the heme group, which acts as the prosthetic moiety of hemoproteins, hence exerting a key biological role. In animals, porphyrins are thus compounds that participate in essential biochemical processes (e.g., hemoglobin, myoglobin and cytochromes), present in internal body fluids and tissues [1]. But under certain pathological conditions (i.e., rare genetic disorders termed porphyrias), porphyrins also accumulate in external structures such as skin [2,3], and some birds and mammals also accumulate them in the integument under physiological conditions [4,5,6,7]. These patterns of occurrence of porphyrins mean, on one hand, that a detailed identification and quantification of porphyrins is essential for the diagnosis of human porphyrias, which is still challenging [2,8]. On the other hand, the quantification of porphyrins in integumentary structures such as feathers and hairs is necessary for a comprehensive understanding of the evolution and ecology of animal form and function.

Several techniques have been used for the study of porphyrins, such as high-performance liquid chromatography (HPLC) [9,10,11,12,13], capillary electrophoresis (CE) [14], spectrofluorometry [15], laser desorption/ionization time of flight mass spectrometry (LDI-TOF–MS) [16] and fast atom bombardment mass spectrometry (FAB-MS) [17]. HPLC is currently established as the standard technique for the quantitative analysis of porphyrins for the diagnosis of human porphyrias [3,5]. HPLC and these other techniques, however, present major disadvantages including long extraction times due to matrix effects [18], a lack of peak separation or focusing, a reduced stability of species that leads to the appearance of new species peaks, irregular retention times [19], and low analytical sensitivity [20].

Here, we develop and validate a new method for the quantification of natural porphyrins in hair using capillary liquid chromatography-mass spectrometry (CLC-MS), which may represent a better alternative to the abovementioned techniques used for porphyrin determination. The a priori advantage of CLC-MS is the requirement of small sample sizes and the achievement of a better sensitivity as compared to other analytical methods. CLC-MS also provides faster measurements with a lower detection limit. Importantly, CLC-MS combines the high selectivity and separation efficiency provided by chromatography with the structural information and the selectivity enhancement provided by mass spectrometry. Furthermore, the coupling of CLC to MS is relatively simple to achieve, thanks to the development of atmospheric pressure ionization sources such as electrospray and atmospheric pressure chemical ionization. These ionization sources are comparatively soft and mainly produce the molecular ions of interest [21,22].

Hair is a very useful material from which to obtain natural porphyrins due to its inert nature and the non-invasive requirement of its collection as compared to body fluids such as blood. It also offers the opportunity to investigate ecological and evolutionary aspects of the animals that deposit porphyrins in integumentary structures, as stated above. One of the mammals in which pelage potentially contains porphyrins, due to the recently discovered pink fluorescence of its hair, is the Southern flying squirrel Glaucomys volans, a small sciurid native to the forests of southeastern North America [23]. We therefore validated our CLC-MS method using Southern flying squirrel hair, determining four natural porphyrins that are relevant for human porphyrias [2,5]: uroporphyrin I (UP I), coproporphyrin I (CP I) and III (CP III), and protoporphyrin IX (PP IX) (Table 1).

**Table 1 ijms-23-06230-t001:** The structures of the four natural porphyrins analyzed in this study.

Compounds	Chemical Structure	Pka	Ref.
Uroporphyrin I (UP I)	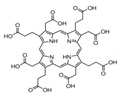	3.16	[24]
Coproporphyrin I (CP I)	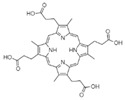	3.56–5.18	[25]
Coproporphyrin III (CP III)	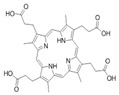	3.56–5.18	[25]
Protoporphyrin IX (PP IX)	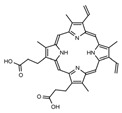	4.94	[26]

## 2. Results

### 2.1. Fluorescence Analysis by UV

Extracts from Southern flying squirrel hairs were irradiated with a Consort nv (Turnhout, Belgium) E2107 UV light with maximum illumination at 365 nm, to confirm the presence of porphyrins on the basis of fluorescence. According to Galván et al. [7], porphyrins in feathers produce intense pink-red fluorescence, and the same was expected here for hairs. All extracts from flying squirrel hair produced fluorescence, showing particularly patent pink coloration in two samples (c and d) (Figure 1). This suggested the presence of porphyrins in hair, with higher concentrations in c and samples.

### 2.2. Optimization of Experimental Parameters

For separation by CLC, we used a chromatographic method consisting in a modification of the procedure described by Mateo et al. [27]. Optimal conditions for the most adequate and efficient LC-MS method to analyze natural porphyrins are shown in Table 2.

Standard solutions of porphyrins (UP I, CP I, CP III and PP IX) were initially analysed by CLC with conventional UV detection at 402 nm, first one by one and then in a mixture of them all, for their identification based on retention time. After multiple injections, the elution order was found, as shown in Figure 2a, to be as follows: UP I was eluted first at 2.5 min, followed by CP I (~8.0 min), CP III (~12.0 min) and PP IX (~14.0 min). Once the analytes were identified it was necessary to quantify them given that conventional UV detection is not sensitive enough for this purpose. Thus, the CLC device was coupled to a single quadrupole MS. Firstly, several analyses of the mixture in scan mode were performed to gather the MS spectra of every single chemical and select the ions to be monitored in SIM mode. These MS spectra are shown in Figure 3. Sensitivity provided by SIM is higher than scan mode, therefore SIM was chosen. Then target ions for all compounds were introduced as MS parameters for SIM mode operation. They were: 831.3 for UP I, 711.2 for CP I, 655.0 for CP III, and 563.3 for PP IX. The single quadrupole MS parameters such as drying gas flow, drying gas temperature, nebulizer pressure, and fragmenting voltage were optimized for each analyte in flow injection (FIA) mode without using a stationary phase column. The optimal experimental conditions are summarised in Table 2, including range assayed.

Using the optimized method, we could fully separate the four target porphyrins (Figure 2). UP I was eluted first, followed by CP I, CP III and PP IX.

### 2.3. Analytical Performance of the Method

Using the optimized method, we conducted a linearity test with porphyrin concentrations at 0.01, 0.025, 0.05, 0.1, 0.25, 0.5, 1, 2, 2.5 and 5 µg mL^−1^. Calibration curves were plotted after six successive injections of each analyte, repeating each injection at least three times. Results are summarized in Table 3. LOD and LOQ were less than 0.2 and 0.7 µg mL^−1^, respectively. Additionally, acceptable within-run and between-run precision values were calculated for the proposed methodology. Repeatability values ranged from 0.2 to 0.5% and from 2 to 4.2% for retention time and peak area, respectively.

These results show that we have developed a validated, efficient and high precision method for the extraction of natural porphyrins from hair.

### 2.4. Application of the Method to Natural Hairs

Once the operating conditions were optimized, the analysis of real samples was performed. First, four samples of Southern flying squirrel hair were prepared as described before porphyrin extraction and were analyzed for their porphyrins content. The resulting LC-MS chromatograms are shown in Figure 4. The back hair contained UP I and PP IX in one of the specimens (Figure 4a), and UP I, CP I and PP IX in the other specimen (Figure 4b). The ventral hair contained the four porphyrins (UP I, CP I, CP III and PP IX) in both specimens. The extracts of these samples were then spiked at four levels (50, 75, 100 and 150 µg/g) with the existing detected porphyrin and the not existing one, in order to evaluate the accuracy and the resolution obtained in the method. Also, the fortification of porphyrins was carried out in order to confirm the presence of these analytes in the analyzed real samples and avoid the false positive results. Table 4 lists the quantitative levels of each porphyrin, as well as the recoveries obtained in each sample, which were estimated from measured amounts versus the added amounts. Resolution was kept under optimum conditions during the analysis of real samples, and acceptable recoveries in the range 90.0–102.7%, were obtained in all cases.

The mass concentrations of the target porphyrins in flying squirrel hair were calculated (Table 4).
(1)$$\mathrm{The}\;\mathrm{theoretical}\;\mathrm{concentrations}=\frac{m}{V_{t}},$$
were 8.8, 11.5, 5.2 and 3.6 g L^−1^ for samples a, b, c and d, respectively.
(2)$$\mathrm{The}\;\mathrm{concentrations}\;\mathrm{found}\;\mathrm{in}\;\mathrm{practice}=\frac{C_{c}}{C_{t}},$$

However, these values varied between 22 and 353.2 µg g^−1^ depending on the presence of each porphyrin in each sample.

With:

m = the mass of flying squirrel hair

V_t_ = the total volume

C_c_ = the concentration calculated from the calibration line

C_t_ = the theoretical concentration.

The yield calculations for the four samples are summarized in Table 5. Recovery values ranged from 86.4% to 98.6%, and the percentages of RSD ranged from 1.5% to 3.3%. These results confirm that our method was successfully applied to determine UP I, CP I, CP III and PP IX in natural hair.

## 3. Discussion

Following the optimization of all previous conditions, the new fixed method was subsequently used to analyze, separate and extract the analytes in the Southern flying squirrel hair samples. After several injections, the order of elution was as follows: UP I (2.821 min), CP I (8.182 min), CP III (11.668 min) and PP IX (13.88 min).

This analysis was done by CLC-MS, which is a revolutionary tool in the chemical and life sciences. LC/MS is accelerating chemical research by providing a robust separation and identification tool for chemists and biologists in diverse fields. LC/MS is best done with capillary HPLC. Capillary HPLC uses smaller column internal diameters than conventional HPLC. Smaller ID columns, for fixed amounts of injected material, produce taller peaks. Taller peaks provide better detection limits for mass spectrometry and other concentration sensitive detectors. For the same amount of material injected, the peak height is inversely proportional to the cross sectional area of the column. The use of smaller ID columns requires careful planning if you are used to normal 4.6 mm columns.

After the identification of the studied analytes, we used the scanning mode to associate each analyte with its MS spectra to use them in SIM mode. By comparing the two modes, we observed that the SIM mode had a higher sensitivity than the scan mode; hence, we used the SIM mode for the rest of the analyses. The target ions were analyzed in a single channel and identified with a time under 15 min. The main ions in the mass spectra for UP I, CP I, CP III and PP IX were 831, 711, 655 and 563, respectively.

We can explain these results by differences in polarity between porphyrins (and the non-polar nature of the stationary phase used (C18)). UP I is the most polar porphyrin, being eluted first, while the least polar compound, PP IX, was retained in the column and eluted last. CP I and CP III are two isomers, thus presenting virtually the same polarity, but the presence of dihydrochloride (2 HCl) in the standard used for CP I increases its polarity, which explains that CP I was eluted second and CP III third.

According to the chromatograms and the calculated porphyrin concentrations, the samples of ventral hair (c and d) contained higher porphyrin concentrations than those of back hair (a and b). This may explain the presence of pink fluorescence observed in c and d (Figure 4). The pink coloration is the result of the structure of the porphyrins, which are composed of four pyrroles linked by methine bridges to form highly conjugated aromatic rings that fluoresce when excited with ultraviolet light (Figure 1). In fact, the exposure to visible light leads to the breakage of rings and the photodegradation of the pigment [28].

Therefore, this study can be applied to determine natural porphyrins in hair and other biological samples, including human samples.

Additionally, the method has allowed us to confirm the presence of porphyrins in the hair of Southern flying squirrels for the first time.

The evolution of visual displays in animals has been a focus of evolutionary biologists since Darwin and Wallace first debated the topic in the nineteenth century (overview in Cronin [29]). In recent years, evolutionary biologists studying color have emphasized a need for the detailed understanding of the pigments and pathways for pigment synthesis that underlie color displays to properly understand the signal function of these displays [30]. Among all pigments used for color displays in animals, the distribution, function, and evolution of integumentary coloration resulting from porphyrins in birds and mammals are among the most poorly understood, primarily because of a lack to techniques for quantifying porphyrins in natural tissue. The new methods that we present will create fantastic new opportunities for the study of porphyrins in natural systems and will advance the understanding of the signal function of porphyrin-based coloration.

## 4. Materials and Methods

### 4.1. Materials

LC-MS grade acetonitrile (ACN) and methanol (MeOH) were purchased from Fisher Scientific (Loughborough, Leics, UK). Hydrochloric acid was purchased from Panreac (Barcelona, Spain). *N*,*N*-Dimethylformamide and formic acid (FA) were obtained from Sigma Aldrich. Deionised water was obtained from a Milli-Q purification system (Millipore, Bedford, MA, USA). Porphyrin standards were purchased as individual compounds from Sigma Aldrich and Frontier Scientific (Table 6).

### 4.2. Sampling of Hair

Two Southern flying squirrel specimens, collected in March 2020 in Bullock County, AL, USA and deposited in the mammal collection at Auburn University (USA), were sampled for hair. These specimens were stored in a dark freezer from death until Ca. Fifty hairs were clipped from the back and the belly of each specimen, deposited in plastic bags and stored in the dark until the analyses were performed.

### 4.3. Instrumentation

A capillary LC pump (Agilent series 1200, Waldbronn, Germany) was used for the chromatographic system, and a C18 reverse phase Luna C18, 5 µ (size 250 × 0.50 mm^2^) column from Supelco (Bellefonte, PA, USA) was used for the chromatographic separation of the analytes. Detection was carried out with a UV–Vis diode array detector (Agilent, 1260 infinity model) equipped with a 2 μL flow cell coupled in series to an Agilent 6110 MS detector (Waldbronn, Germany) equipped with an atmospheric pressure ionization source electrospray (API-ES). In addition, a V-630 UV-Vis spectrophotometer equipped with a 10-mm flow cell holder was used to measure the wavelengths of the four studied porphyrins.

### 4.4. Standard Preparation and Sample Treatment Procedures

The stock solutions of all porphyrins were prepared at 1.0 mg mL^−1^ in 50% MeOH and 50% *N*,*N*-dimethylformamide and stored in darkness at −4 °C. Working standard solutions were prepared at 0.1 mg mL^−1^ by appropriate dilution in 50% ACN and 50% 6 M FA. Standard calibration samples were prepared by appropriately diluting the working standard solution of each analyte.

The procedure for extracting porphyrins from Southern flying squirrel hairs consisted in several steps following a protocol modified from Mateo et al. [28], performing the whole process the same day. First, hairs were weighed and trimmed, and placed in a Falcon tube. Then, the same volume of 6 M FA and ACN was added to each tube, shaking to immerse all hairs in the liquid. The tubes were incubated for 24 h in the dark after 15 min of cold sonication. The liquid was then extracted using Pasteur pipettes, deposited in tubes and centrifuged for 5 min at 10,000× *g* and 4 °C (sediments were not appreciated). Supernatants were deposited in LC-MS vials through plastic round filters for CLC-MS analyses, which included calibration curves with UP I, CP I, CP III and PP IX standards.

### 4.5. Separation of Porphyrins by LC/MS and Operating Conditions

We used a chromatographic method consisting in a modification of the procedure described by Mateo et al. [27]. The analytes were separated by a 16 min gradient elution using a three component mobile phase consisting of solvent A (0.2% FA in water) and solvent B (50% ACN + 50% MeOH). The gradient elution started with the injection of 50% B up to 3 min, then increased to 95% B in 12 min and then returned to 50% B in 1 min. The column was allowed to re-equilibrate for 5 min at 50% B before the next injection. The detection of porphyrins was made in positive electrospray ionization mode (ESI+). MS parameters were as follows: drying gas flow 5 L/min, gain 10, fragmentor 350, ion source temperature 200 °C, capillary voltage 5 kV and nebulizer gas (N2) 35 psi. The full analysis mode was used to identify analytes by matching retention time and mass spectra to standards, while the single ion monitoring (SIM) mode was used to quantify the target analytes.

### 4.6. Spectrofluorometric Analysis

To confirm the presence of fluorescence, the four target extracts were measured using a Photon Technology International Inc. Quanta Master 40 spectrofluorometer equipped with a photomultiplier and a graphing spectrophotometer. The equipment was zeroed against a solution of 50% ACN and 50% 6 M FA at an excitation wavelength of 365 nm and an emission wavelength of 625 nm, with the monochromator slit width set at 1 nm. The emission spectrum was then scanned from 600 to 650 nm. Results confirmed the fluorescence visually perceived in Figure 1, showing highest fluorescence intensities in c and d samples (Figure 5).

## 5. Conclusions

The present study allowed the development of a new analytical method of LC-MS for the simultaneous determination of natural porphyrins (UP I, CP I, CP III and PP IX) in mammalian hair. This method presents several advantages to previous chromatographic methods used for porphyrin analysis such as simplicity, high precision, selectivity, and sensitivity.

The procedure to extract porphyrins from hair, by using 6 M FA/ACN, incubation and sonication processes, was easy, fast and inexpensive, while requiring low processing volumes and generating reliable results with better extraction yields.

To rule out false positives or possible interferents, as typically proposed by using single quadrupole mass spectrometry as detection in the CLC, extracts of the analyzed samples were fortified at four levels (50, 75, 100 and 150 µg/g), confirming their good selectivity.

The method can therefore be applied to determine the natural porphyrins in hair and other biological samples, including human samples. Our proposed method therefore has the potential to become the standard technique for porphyrin analysis, which may thus facilitate the early diagnosis of porphyrias given the analytical simplicity and speed [2,3,5]. Additionally, the method has allowed us to confirm the presence of porphyrins in the hair of Southern flying squirrels for the first time. The distribution, function, and evolution of integumentary coloration resulting from porphyrins in birds and mammals are poorly understood, primarily because of a lack to techniques for quantifying porphyrins in natural tissue. The new methods that we present will create fantastic new opportunities for the study of porphyrins in natural systems.

## Figures and Tables

**Figure 1 ijms-23-06230-f001:**
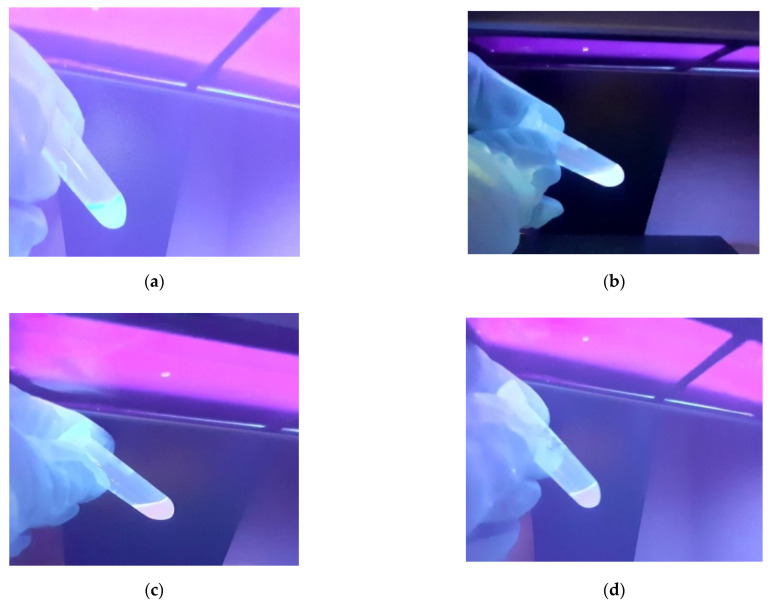
Fluorescence observed in Southern flying squirrel hair extracts under UV light, showing pink coloration in (**c**,**d**) samples. Samples (**a**,**b**) correspond to back hair, while samples (**c**,**d**) correspond to ventral hair.

**Figure 2 ijms-23-06230-f002:**
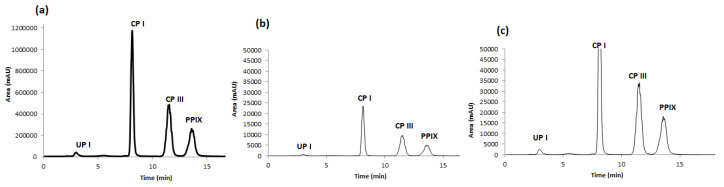
LC-MS chromatograms of: (**a**) standard solution of the porphyrin mixture at 10 μg mL^−1^, (**b**) LOD of the porphyrin mixture at 0.2 μg mL^−1^ and (**c**) LOQ of the porphyrin mixture at 0.7 μg mL^−1^. Conditions: stationary phase Luna C18, 5 µm (size 250 × 0.50 mm^2^); mobile phase A = 0.2% formic acid in water/B = 50% ACN + 50% MeOH; flow rate 15 μL min^−1^; injection volume 5 μL; detection at 402 nm.

**Figure 3 ijms-23-06230-f003:**
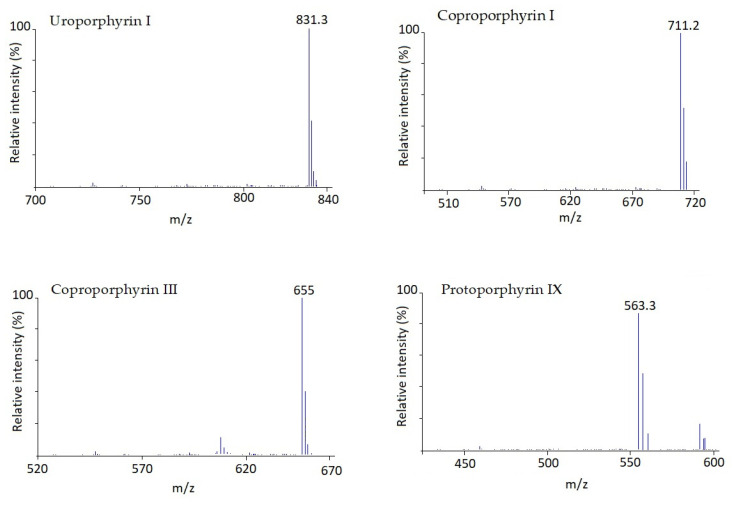
Ion spectrum of Uroporphyrin I, Coproporphyrin I, Coproporphyrin III and Protoporphyrin IX.

**Figure 4 ijms-23-06230-f004:**
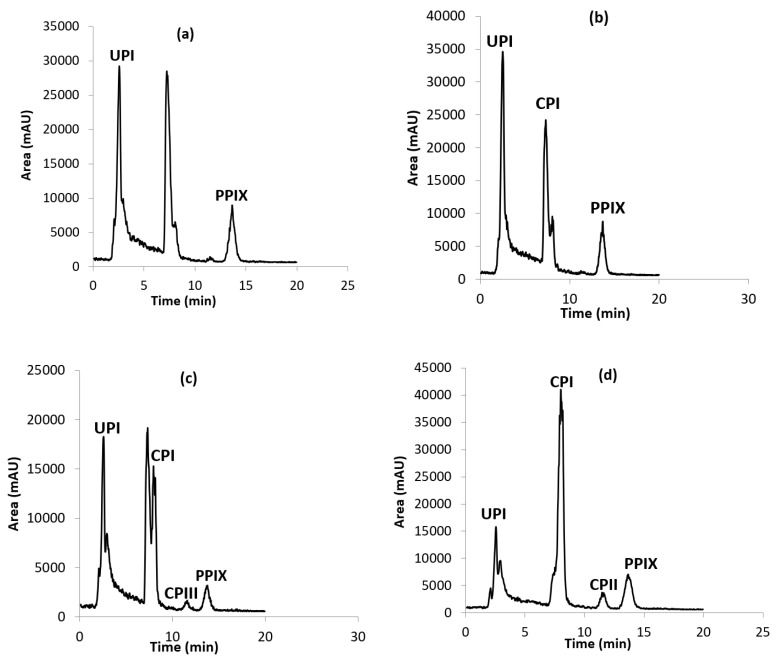
LC-MS chromatograms of four hair samples (**a**–**d**) from two Southern flying squirrel specimens. Samples (**a**,**b**) are from back hair and samples c and d are from ventral hair. Conditions: stationary phase Luna C18, 5 µm (size 250 × 0.50 mm^2^); mobile phase A = 0.2% Formic acid in water/B = 50% ACN + 50% MeOH; flow rate 15 μL min^−1^; injection volume 5 μL; detection at 402 nm.

**Figure 5 ijms-23-06230-f005:**
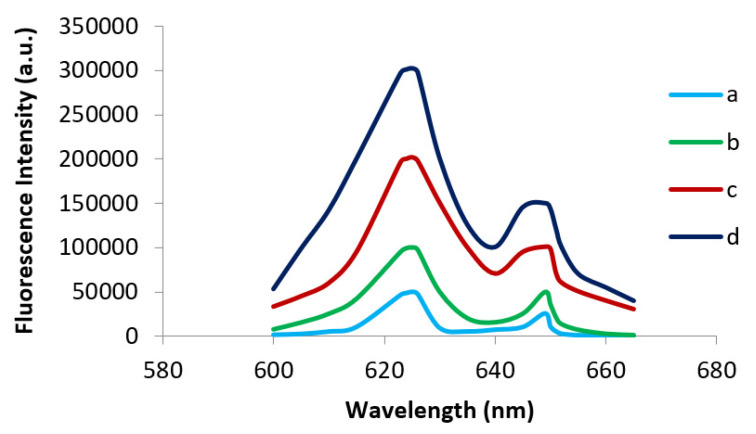
Overlay of emission spectra of four samples of Southern flying squirrel hair (a–d). λ_em_ = 625 nm and λ_ex_= 365 nm.

**Table 2 ijms-23-06230-t002:** Optimised operating conditions for the determination of natural porphyrins in hair by CLC-MS.

Parameters of Chromatographic Method	Range Assayed	Optimal Conditions
Stationary phase	Ascentis C18 and Luna C18	Luna C18
Mobile phase composition	Water, FA, ACN and MeOH	0.2% FA/50% ACN + 50% MeOH
Elution Mode	Isocratic and gradient	Gradient
Injection volume (µL)	2–5	5
Flow rate (µL min^−1^)	10–20	15
Drying gas flow (L min^−1^)	5–13	5
Drying gas temperature (°C)	150–250	200
Nebulizer pressure (Psi)	30–40	35
Capillary voltage (v)	4000–5500	5000

**Table 3 ijms-23-06230-t003:** Linearity, calibration data and figures calculated for the chromatographic method proposed for the analysis of natural porphyrins in hairs.

Analyte	Linear Range (µg mL^−1^)	Linearity Curve	R^2^	S_y/x_	Repeatability, RSD%		LOD (µg mL^−1^)	LOQ (µg mL^−1^)
					t_R_	Peak Area		
UP I	0.1–5	y = 537.27x + 10697	0.998	55.96	0.2	2.0	0.199	0.665
CP I	0.01–2	y = 1175.7x − 26862	0.997	60.49	0.5	4.2	0.097	0.323
CP III	0.01–1	y = 872.92x + 2165.2	0.999	3.14	0.3	2.8	0.006	0.021
PPIX	0.05–1	y = 440.49x + 3604.5	0.999	3.67	0.4	3.4	0.016	0.055

**Table 4 ijms-23-06230-t004:** Porphyrin concentrations in extracts of Southern flying squirrel hair.

Hair Samples	Analyte	Found (µg/g)	Added (µg/g)	Found (µg/g)	Recoveries (%)
a	UP I	254.0	50	45.0	90.0
	CP I	ND	75	72.0	96.0
	CP III	ND	100	95.0	95.0
	PP IX	56.0	150	140.0	93.3
b	UP I	229.0	50	47.0	94.0
	CP I	22.0	75	76.0	101.3
	CP III	ND	100	98.0	98.0
	PP IX	43.1	150	142.0	94.7
c	UP I	300.3	50	44.0	88.0
	CP I	145.6	75	77.0	102.7
	CP III	15.7	100	95.0	95.0
	PP IX	52.3	150	145.0	96.7
d	UP I	353.2	50	49.0	98.0
	CP I	270.0	75	70.0	93.3
	CP III	27.5	100	97.0	97.0
	PP IX	133.0	150	146.0	97.3

ND: Not detected.

**Table 5 ijms-23-06230-t005:** Recoveries and RSD values of porphyrins in Southern flying squirrel hair.

Porphyrin	Recoveries %				RSD %
	a	b	c	d	
UP I	97.1	97.6	96.4	97.8	1.5
CP I	ND	96.6	94.6	97.3	2.2
CP III	ND	ND	90.3	98.6	2.6
PP IX	98.6	86.4	90.6	95.7	3.3

ND: Not detected.

**Table 6 ijms-23-06230-t006:** Porphyrin reagents used to prepare standard porphyrin solutions in the study.

Compound	Abbreviation	CAS No.	Purity (%)	Mw (Da)	Manufacturer
Uroporphyrin I	UP I	607-14-7	≥90	830.75	Frontier Scientific Inc.
Coproporphyrin I	CP I	69477-27-6	≥90	727.63	Sigma Aldrich
Coproporphyrin III	CP III	14643-66-4	≥97	654.71	Frontier Scientific Inc.
Protoporphyrin IX	PP IX	553-12-8	≥95	562.66	Sigma Aldrich

## Data Availability

The data presented in this study are available in this article.

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
