# Peer review of "Detection of Porphyrins in Hair Using Capillary Liquid Chromatography-Mass Spectrometry"

_ijms, 2022, doi:10.3390/ijms23116230_

Round 1

Reviewer 1 Report

According to the paper, the single ion monitoring mode was used to quantify the target analytes. The spectra and chromatogram of each should be provided in the manuscript. Besides, the chromatogram of the LOD and LOQ should also be added in the figures.

In the experiment section, the method of sample treatment and preparation of the standard solution should be added.

The methodological investigations, such as stability, should be provided in this manuscript.

In the abstract, line 25,” using detection wavelength at 402 nm, UV fluorescence measurements at 365 nm”. But according to the paper, the SIM was used in the paper. Therefore, this sentence in the abstract needs to be deleted.

In the figure 3, the chromatograms of the four hair sample could be used to quantitative analysis. Baseline anomalies could lead to problems with identification and quantitation of analytes.

Author Response

The authors thank the  reviewers for suggestions and comments, which have contributed to improve the overall quality of the new manuscript. We introduced changes in the new version accordingly. Some comments are included below:

  • LC-MS chromatograms of LOD and LOQ and ion spectrums were added in the new version of the manuscript
  • In the experiment section, the method of sample treatment and preparation of the standard solution have been added in the new version of the manuscript. See section 4.4. in the new version of the manuscript.
  • Sentence “using detection wavelength at 402 nm, UV fluorescence measurements at 365 nm and spectrofluorometer with emission at 625 nm” has been deleted.
  • Quantitative results of porphyrins in the four hair samples have been expressed in Table 3 of the new version of the manuscript.
  • The anomalies present in the baseline of these chromatograms are very small compared to the intensities of the peaks found; therefore, they cannot affect the results of quantitative measurements for the analytes studied.

Reviewer 2 Report

The manuscript may be a valuable contribution but too little data is presented to justify the claims made. A major revision is suggested.

Author Response

The authors thank the  reviewer for suggestions and comments, which have contributed to improve the overall quality of the new manuscript. We introduced changes in the new version accordingly. Some comments are included below:

  • Variables involved in LC-MS analyses were ptimized, namely stationary phase, mobile phase composition, elution mode, injection volume, flow rate, drying gas flow, drying gas temperature, nebulizer pressure, and capillary voltage. The optimal experimental conditions are summarized on Table 1, including variables assayed in certain ranges. These optimal conditions gave the best results in terms of analysis, separation and identification of the studied compounds. Once all these conditions were fixed, the same procedure was followed for the analysis of all replicates with the aim to extract, separate and then identify the analytes according to retention time by means of LC-MS analyses.
  • In order to evaluate the simultaneous detection of four studied porphyrins, as well as their identification, every sample was spiked with the existing detected analyte and the not existing one. Results have been added in the Table 3 and ion chromatograms of studied porphyrins were represented in Figure 3 of the new version of the manuscript.
  • The fluorescence images were changed at the beginning to confirm the initial hypothesis that the hair might contain porphyrins.
  • The theoretical concentrations were calculated by the following equation:

The theoretical concentrations =

With:

m = the mass of each flying squirrel hair (a, b, c and d)

Vt = the total volume (2 mL = 1 mL of 6M FA + 1 mL of ACN)

Sample

a

b

c

d

m (g)

0.0176

0.023

0.0104

0.0072

Vt (L)

0.002

0.002

0.002

0.002

Theoretical concentration

8.8

11.5

5.2

3.6

  • This analysis was done by CLC-MS which is a revolutionary tool in the chemical and life sciences. LC/MS is accelerating chemical research by providing a robust separations and identification tool for chemists and biologists in diverse fields. LC/MS is best done with capillary HPLC. Capillary HPLC uses smaller column internal diameters than conventional HPLC. Smaller ID columns, for fixed amounts of injected material, produce taller peaks. Taller peaks provide better detection limits for mass spectrometry and other concentration sensitive detectors. For the same amount of material injected, the peak height is inversely proportional to the cross-sectional area of the column. The use of smaller ID columns requires careful planning if you are used to normal 4.6 mm columns. This information was added in the new version of the manuscript.

Round 2

Reviewer 1 Report

Accept in present form.

Author Response

The authors thank the reviewer which has contributed to improve the overall quality of our manuscript.

Reviewer 2 Report

n Table 2 the regression equations use a comma as a delimiter other values use a period.  I suggest using a period.

In the paragraph starting on line 174, it seems you are using a method of standard addition to estimate recoveries.  The "found" values were apparently determined in non-spiked samples and the spiked recoveries were determined by difference. These are significant details and should be more clearly presented. 

Table 1 summarizes the conditions you evaluated while developing the method. You should add a statement that not all details of the validation are presented but the best method based on a specific criteria which you must specify in the manuscript should be included.

Table 3 presents recoveries in unspiked samples and this should be clarified for a reader. The values are calculated based on spike recoveries.

Line 318 should be reworded to include that samples were spiked at target concentrations for the 4 standards. Specify target conc.  Both spiked and unspiked sample prep should be clearly described. 

Manuscript is improved from the earlier draft but still could be improved by providing more details about what was done and how the data were handled.

Author Response

Comment: Table 2 the regression equations use a comma as delimiter other values use a period.  I suggest using a period.

Answer: Sorry for this mistake, period has been used in the new version of the manuscript.

Comment: In the paragraph starting on line 174, it seems you are using a method of standard addition to estimate recoveries.  The "found" values were apparently determined in non-spiked samples and the spiked recoveries were determined by difference. These are significant details and should be more clearly presented. 

Answer: This paragraph has been clarified in the new version of the manuscript.

Comment: Table 1 summarizes the conditions you evaluated while developing the method. You should add a statement that not all details of the validation are presented but the best method based on a specific criteria which you must specify in the manuscript should be included.

Answer: For separation by CLC, we used a chromatographic method consisting in a modification of the procedure described by Mateo et al [30]. This information has been included in the new version of the manuscript. Information about optimization of the single quadrupole MS parameters have been also included in the new version of the manuscript.

Comment: Table 3 presents recoveries in unspiked samples and this should be clarified for a reader. The values are calculated based on spike recoveries.

Answer:This comment has been taken into account in the new version of the manuscript.

Comment: Line 318 should be reworded to include that samples were spiked at target concentrations for the 4 standards. Specify target conc.  Both spiked and unspiked sample prep should be clearly described.

Answer: This information has been included in conclusions section of the new version of the manuscript. More details about fortification process and sample treatment can be also found in sections 2.4 and 4.4.

Comment: Manuscript is improved from the earlier draft but still could be improved by providing more details about what was done and how the data were handled.

Answer: All these comments have been taken into account. The authors thank the reviewer for these comments, which have contributed to improve the overall quality of the new manuscript.
